# Ca^2+^-Dependent Processes of Innate Immunity in IBD

**DOI:** 10.3390/cells13131079

**Published:** 2024-06-21

**Authors:** Francesco Palestra, Gina Memoli, Annagioia Ventrici, Marialuisa Trocchia, Mariarosaria Galdiero, Gilda Varricchi, Stefania Loffredo

**Affiliations:** 1Department of Translational Medical Sciences, University of Naples Federico II, 80131 Naples, Italy; f.palestra97@gmail.com (F.P.); gina.memoli98@gmail.com (G.M.); annagioia.v99@gmail.com (A.V.); marytrocchia@gmail.com (M.T.); mrgaldiero@libero.it (M.G.); gildanet@gmail.com (G.V.); 2Center for Basic and Clinical Immunology Research (CISI), University of Naples Federico II, 80131 Naples, Italy; 3World Allergy Organization (WAO), Center of Excellence, 80131 Naples, Italy; 4Institute of Experimental Endocrinology and Oncology, National Research Council (CNR), 80131 Naples, Italy

**Keywords:** inflammatory bowel disease (IBD), ulcerative colitis, Crohn’s disease, monocytes/macrophages, dendritic cells, mast cells, granulocytes, cytokines, Ca^2+^-signalling

## Abstract

IBD is an uncontrolled inflammatory condition of the gastrointestinal tract, which mainly manifests in two forms: ulcerative colitis (UC) and Crohn’s disease (CD). The pathogenesis of IBD appears to be associated with an abnormal response of innate and adaptive immune cells. Innate immunity cells, such as macrophages, mast cells, and granulocytes, can produce proinflammatory (e.g., TNF-α) and oxidative stress (ROS) mediators promoting intestinal damage, and their abnormal responses can induce an imbalance in adaptive immunity, leading to the production of inflammatory cytokines that increase innate immune damage, abate intestinal barrier functions, and aggravate inflammation. Considering that Ca^2+^ signalling plays a key role in a plethora of cellular functions, this review has the purpose of deepening the potential Ca^2+^ involvement in IBD pathogenesis.

## 1. Introduction

The growing prevalence of inflammatory bowel disease (IBD), which affects about 5 million people worldwide, has led to a need for novel diagnostic and therapeutic approaches to treat IBD patients [1].

IBD is an uncontrolled inflammatory condition of the gastrointestinal tract, which mainly manifests in two forms: ulcerative colitis (UC) and Crohn’s disease (CD), both of which are characterized by chronic inflammation in the gastrointestinal tract and repeated cycles of relapse and remission. The pathogenesis of IBD appears to be associated with abnormal gut microbiota, environmental changes, gene variants, and immune response dysregulation [2,3,4,5].

Even though IBD aetiology and pathogenesis are not completely understood, evidence suggests that the pathology is driven by an abnormal activation of immune cells in the mucosa of IBD patients [6,7]. Both innate and adaptive immune cells play key roles in the pathomechanism of IBD [4,6,8,9,10]. Innate immunity cells, such as macrophages, mast cells, and granulocytes, can produce proinflammatory (e.g., TNF-α) and oxidative stress (ROS) mediators promoting intestinal damage, and their abnormal responses can induce an imbalance in adaptive immunity, leading to the production of inflammatory cytokines that increase innate immune damage, abate intestinal barrier functions, and aggravate inflammation [4,8,11]. This creates a vicious cycle that contributes to the development of IBD. Adaptive immune cells (e.g., T cells, B cells, and regulatory T cells), are also involved in the pathomechanism of IBD [10]. For example, Th17 cells and IL-17 are overproduced in IBD patients, while the number of regulatory T cells is reduced [6,12].

Blockade of these pro-inflammatory mediators produced by immune cells could be a mechanism to alleviate or even reverse symptoms of IBD. In fact, recent advances that have revolutionized IBD treatment include blocking antibodies against pro-inflammatory cytokines such as TNF-α (e.g., infliximab or adalimumab), IL-12, and IL-23 (ustekinumab) [10]. Similarly, targeting the migration of these immune cells to the gut, by blocking integrin function, has also shown remarkable efficacy and this method is commonly used in IBD management. Vedolizumab has been the first anti-adhesion drug clinically approved for IBD treatment [10,13]. Other anti-adhesion molecules (e.g., etrolizumab, PN-943, and AJM300) are currently in preclinical and clinical trials [13].

Calcium (Ca^2+^) is essential in regulating a plethora of cellular functions that include cell proliferation and differentiation, axonal guidance and cell migration, neuro/enzyme secretion and exocytosis, development/maintenance of neural circuits, cell death, and many more [10]. Although Ca^2+^ is fundamental in physiological signalling, its dysregulation is often involved in pathological processes such as Alzheimer’s disease, Huntington’s disease, Parkinson’s disease, and dystrophies [14,15]. Ca^2+^ dysregulation also seems to be involved in the IBD pathomechanism [10,16,17,18].

Considering that Ca^2+^ signals play a crucial role in immune system function [19,20], such as in cytokine and chemokines secretion, in this review, we discuss the potential involvement of immune cells and their Ca^2+^-dependent processes in IBD pathogenesis and the possibility to modulate this signalling as a therapeutic avenue.

## 2. Innate Immunity in IBD

The intestinal immune system must strike a balance between preventing invasion and tissue damage produced by inflammation to maintain homeostasis and guarantee optimal energy intake. This imbalance can result in IBD arising [21]. As above mentioned, IBD consists of two separate disorders with differing clinical manifestations: UC is an ulcerating mucosal disease that mostly affects the colon; CD is a deep transmural inflammation that mostly affects the terminal ileum but can affect any part of the body from the mouth to the anus. Both diseases are characterized by immunological disorder; particularly, both innate and adaptive immune cells are involved in IBD pathogenesis [10,21]. 

The immune system can be divided into two components: innate and adaptive immunity. The synergistic collaboration between innate and adaptive immune responses is crucial for maintaining immune homeostasis and protecting the host from diverse challenges, including allergens, toxins, malignant cells, and infectious agents such as bacteria and viruses [22]. Monocytes, macrophages, dendritic cells, mast cells, basophils, neutrophils, eosinophils, and natural killer cells comprise the innate immune system, whereas B and T cells comprise the adaptive immune system [22]. Ca^2+^ signalling is required for the maintenance of several immune cell functions, and different investigations have discovered that Ca^2+^ dysregulation is connected to the development of autoimmune and inflammatory disorders [19]. Figure 1 show the mechanisms of innate immunity cell involved in IBD pathogenesis.

### 2.1. Monocytes

Monocytes are a type of white blood cell that are part of the innate immune system. They are produced in the bone marrow and circulate in the bloodstream, where they can migrate to tissues and differentiate into macrophages or dendritic cells. Monocytes are essential effector cells in the innate immune response, exhibiting multifaceted roles in host defence. Their functions encompass phagocytosis and the subsequent degradation of pathogens, antigen processing and presentation to adaptive immune cells, and the secretion of effector molecules such as chemokines and cytokines, which orchestrate inflammatory responses. Furthermore, monocytes are considered to be the primary precursors for intestinal macrophages, highlighting their importance in maintaining gut homeostasis and immunity [8].

In humans, three subsets of monocytes are described in the blood with seemingly different functions [8]:Classical (CD14++CD16−) monocytes are large granular cells with low susceptibility to undergo apoptosis. They are the most abundant subset of monocytes in the blood and are involved in phagocytosis, antigen presentation, and cytokine production;Intermediate (CD14++CD16+) monocytes have intermediate levels of CD14 and CD16 expression and are involved in antigen presentation, cytokine production, and wound healing;Nonclassical (CD14+CD16++) monocytes have high levels of CD16 expression and low levels of CD14 expression. They are involved in patrolling the endothelium for signs of infection or damage, as well as in tissue repair and regeneration. These subsets have seemingly distinct functions within homeostasis and inflammation based on receptor expression, gene-expression profiles, and cytokine responses.

Peripheral blood monocyte populations exhibit notable alterations in patients with CD and UC. Studies have consistently reported a decrease in the proportion of classical monocytes in circulation, suggesting their recruitment to the inflamed intestinal mucosa where they contribute to inflammatory processes. Conversely, CD16+ monocytes are found to be elevated in the peripheral blood of CD patients compared to healthy individuals, indicating a potential role for this monocyte subset in the systemic manifestations of CD [8]. Furthermore, IBD patients with monocytosis are statistically at increased risk for worse clinical outcomes, hospitalization, surgery, and emergency department use [23].

### 2.2. Macrophages

Macrophages are white blood cells located in tissues and are derived from monocytes; they play a significant role in maintaining tissue homeostasis, regulating inflammation, and protecting the host from pathogens. Macrophages can phagocytize and digest foreign substances and can acquire pro-inflammatory (M1-like) or anti-inflammatory (M2-like) phenotypes according to their microenvironment [4].

In the intestine, macrophages can be divided into resident and inflammatory macrophages that are distinguished by low or high expression of CD14, respectively [8]. In the acute phase of IBD, the number of macrophages in the gut mucosa grows significantly. It has been found that TREM-1 expression, a transmembrane protein that induces the release of pro-inflammatory factors from macrophages, is low in normal intestinal tissue but increases in an experimental colitis model and in IBD patients, indicating that an abnormal innate immunity response, driven by macrophages, plays a key role in IBD pathogenesis [24].

Macrophages may have a complex role in IBD: on one hand, they are involved in the clearance of pathogens and debris from the gut lumen, which is essential for maintaining its homeostasis; on the other hand, they can produce pro-inflammatory cytokines such as TNF-α, IL-1β, and IL-6 that contribute to the pathogenesis of IBD [6,9]. The excessive release of these pro-inflammatory cytokines impairs the expression and localization of tight junction proteins, leading to increased intestinal epithelial permeability [8,9]. Additionally, macrophages produce reactive oxygen species (ROS) and nitric oxide (NO), which can damage the intestinal epithelium and further impair intestinal barrier function [25]. Heightened intestinal epithelial permeability facilitates the passage of luminal antigens and pathogens into the lamina propria. This influx triggers immune cell activation and the release of inflammatory cytokines, further amplifying intestinal inflammation. In addition, recent studies have shown that unique subsets of macrophages in the intestinal mucosa contribute to the pathogenesis of Crohn’s disease via the IL-23/IFN-γ axis and increased Th17-inducing activity; particularly, IL-23, released by macrophages, activates signal transducer and activator of transcription (STAT-) 4 signalling in memory T lymphocytes, driving IFN-γ production. This initiates a pro-inflammatory loop, as IFN-γ further stimulates innate immune cells to release inflammatory cytokines, amplifying gut inflammation [4].

There are several potential therapeutic approaches targeting macrophages in gut inflammation. For example, anti-TNF therapy promotes apoptosis in intestinal CD14+ macrophages, effectively reducing their numbers within the inflamed gut. Anti-α4β7 therapy, on the other hand, disrupts the trafficking of myeloid cells, including monocytes and macrophages, to the intestine, thereby limiting their contribution to inflammation. IL-12 and IL-23 produced by myeloid cells are blocked with anti-IL-12/23 therapy, influencing Th1 cell- and Th17 cell-response development. Thiopurines reduce the activation of macrophages via GTPase Rac inhibition, while methotrexate has an antiproliferative outcome mediated by apoptosis and an anti-inflammatory effect [8]. Another potential therapeutic approach consists of increasing anti-inflammatory (M2-like) macrophages by promoting the effects of IL-4, which induces M2 polarization. This can be accomplished using IL-4 delivery systems such as nanomaterials and biomaterials. Additionally, some studies have shown that certain metabolites, such as arginase and polyamines, can also promote M2 polarization. Another way to reprogram macrophages is by using short-chain fatty acids (SCFAs), which are produced by the gut microbiota and have been shown to promote M2 polarization [4].

Another potential therapeutic strategy could be represented by targeting specific signalling pathways in macrophages, such as the JAK-STAT pathway. The JAK-STAT pathway is involved in the regulation of cytokine signalling and has been shown to play a role in the pathogenesis of IBD. Inhibition of this pathway may reduce inflammation and promote tissue repair. Some studies have shown that targeting JAK-STAT signalling in macrophages can reduce inflammation and improve symptoms in animal models of IBD [4]. However, further research is needed to fully understand the effectiveness and safety of these therapeutic approaches.

### 2.3. Dendritic Cell

Dendritic cells (DCs) are a type of immune cell that play a crucial role in the initiation and regulation of immune responses. They are found in tissues throughout the body, including the skin, lymph nodes, and mucosal surfaces such as the intestine. DCs are responsible for sampling antigens from their environment and presenting them to T cells, which are key players in the adaptive immune response [21].

Dendritic cells, crucial for bridging innate and adaptive immunity in the mucosa, are likely implicated in the immune pathogenesis of IBD [26]. In fact, in the context of IBD, DCs are known for collecting luminal antigens and delivering them to T cells in the mesenteric lymph nodes, where they activate effector T cells. Furthermore, DCs secrete cytokines that recruit other immune cells to the site of infection or inflammation, such as IL-12 (promotes the differentiation of Th1 cells, which produce IFN-γ and are involved in the response to intracellular pathogens), IL-23 (promotes the differentiation of Th17 cells, which produce IL-17A and IL-17F and are involved in the response to extracellular bacteria and fungi), and TNF-α (promotes the recruitment of immune cells to the site of infection or inflammation and is involved in the regulation of immune responses and the induction of inflammation). These cytokines, particularly IL-17A and IL-17F, have been linked to the damage of the intestinal epithelial barrier [21].

In addition to these cytokines, DCs from IBD patients have been shown to produce increased levels of chemokines such as CCL20 and CXCL1, which recruit immune cells to the site of infection or inflammation [21].

There are two separate subsets of DCs with opposing functions in IBD: the CD103+ DC subset is involved in generating Treg responses, whereas the CX3CR1+ group is more inflammatory. Human studies have shown that DCs accumulate in IBD with higher levels of expression of Toll-like receptor 2 (TLR2), TLR4, and costimulatory molecules, which produced more cytokines than healthy controls. [21]. Furthermore, DCs seem to play a role in regulating the response to gut microflora. In healthy people, CD103+ dendritic cells help to maintain mucosal immune hyporesponsiveness, which is important for preventing inflammation. While DCs typically promote mucosal homeostasis in a hyporesponsive state, IBD patients exhibit an abnormal activation profile of DCs at inflamed intestinal sites. These activated DCs display an enhanced TLR responsiveness, distinguishing them phenotypically from their homeostatic counterparts [26]. Targeting these activated DCs presents a promising therapeutic avenue, though a deeper understanding of DC biology in IBD is crucial for developing effective strategies.

In summary, dendritic cells play a crucial role in the pathogenesis of IBD by sampling luminal antigens and presenting them to T cells, secreting cytokines and chemokines that recruit other immune cells to the site of infection or inflammation, and regulating intestinal epithelial barrier function.

### 2.4. Mast Cells

Mast cells (MCs) are immune cells involved in allergic reactions and inflammation. In IBD, mast cells play central roles in several aspects of the disease, including regulation of epithelium permeability, transmittance of signals during neuropathologic stress, initiation and maintenance of inflammatory responses, and subsequent tissue remodelling that occurs after resolution of the acute inflammatory stage in the gastrointestinal tract [27]. These effects are attributable to mast cells activation by various stimuli such as food antigens, bacteria, and stress, leading to the release of pro-inflammatory mediators such as histamine (HA), prostaglandin D2 (PGD2), leukotriene C4 (LTC4), and cytokines. These mediators can contribute to the recruitment of other immune cells to the site of inflammation and promote tissue damage. In addition, mast cells can also interact with other immune cells such as T cells and dendritic cells to modulate the immune response [6].

The role of MCs in IBD is complex and can be both detrimental and positive. While MCs are known to contribute to inflammation and tissue damage in IBD, they also have a protective role in the gut. Mast cells can release mediators that promote epithelial barrier function and prevent bacterial translocation. Additionally, mast cells can regulate T cell responses and promote regulatory T cell differentiation, which can help to suppress inflammation. Therefore, while mast cells are involved in the pathogenesis of IBD, they also have a protective role in maintaining gut homeostasis [28].

Preclinical studies have shown that mast cell stabilizers, such as cromolyn sodium and ketotifen, can reduce inflammation and improve symptoms in animal models of IBD [29,30,31]. Other potential therapeutic approaches include inhibition of MCs, the inhibitory receptor LIMR3, protease and tryptase inhibitors, or targeting histamine receptors. However, more research is needed to determine the safety and efficacy of these treatments in humans with IBD [32].

Intestinal homing of mast cell progenitors is mainly mediated by the interaction of α4β7 integrin expressed by mast cells with ICAM-1, V-CAM, or MAdCAM-1 expressed by endothelial cells. The positive response of IBD patients to therapy with Vedolizumab, which is a monoclonal antibody that targets α4β7 integrin, suggests that this integrin plays a crucial role in the pathogenesis of IBD. Therefore, targeting α4β7 integrin could be a potential therapeutic strategy for treating gastrointestinal disorders associated with mast cell activation [33].

As mentioned above, the gut microbiota appears to be associated to IBD pathogenesis [4,5,34]. Different studies demonstrate that MCs can modulate the mutual influence between the host and its microbiota through changes in their activation state. Over the last two decades, several reports were able to demonstrate that bacteria and fungi induce mast cell activation. Interestingly, although some microorganisms can elicit a proinflammatory response of MCs, other microorganisms are able to reduce their activation, thus limiting inflammation and promoting homeostatic conditions. Dysregulation of this relationship can lead to the development or exacerbation of gastrointestinal (GI) disorders such as inflammatory bowel diseases (IBD) [33].

The use of probiotics for treating inflammatory bowel diseases, including Crohn’s disease and ulcerative colitis, is supported with increasing scientific evidence. A recent study demonstrated that dietary supplementation with S. cerevisiae-derived β-glucan effectively reduced mast cell-induced intestinal hyperpermeability in both CD patients with ileitis and individuals with non-inflammatory IBD. This finding underscores the significant role of mast cells in the pathogenesis of IBD, particularly in the context of dysbiosis and the resulting intestinal barrier dysfunction. The study provides further support for the therapeutic potential of targeting mast cells and modulating the gut microbiome in managing IBD [33].

### 2.5. Granulocyte Role in IBD

#### 2.5.1. Neutrophils

Neutrophils are a type of granulocyte that plays an important role in the immune response to infection and inflammation [35].

In IBD, neutrophils have been found to be dysregulated and may contribute to the chronic inflammation seen in these conditions. Faecal calprotectin (CALPR), a calcium- and zinc-binding protein used for IBD diagnosis, monitoring, and treatment guidance, is largely produced by neutrophils; in fact, neutrophil migration in the gastrointestinal tract has been shown to be associated with increasing levels of faecal calprotectin in IBD patients [36]. Emerging evidence suggests that faecal neutrophil elastase (NE) activity could be a valuable marker for predicting active ulcerative colitis, potentially rivalling the current gold standard, faecal calprotectin. The two markers reflect different stages of neutrophil activity: calprotectin is released early in inflammation and does not directly damage tissue; by contrast, NE is released later, primarily from damaged or dying neutrophils, and actively contributes to tissue damage at inflamed sites [37]. 

The exact role of neutrophils in IBD is complex and likely depends on many factors, including the stage of disease and the specific subtype of neutrophil involved. In fact, neutrophils seem to have a dual role in intestinal inflammation; on one hand, they may produce pro-inflammatory cytokines such as IL-1β, TNF-α, and CXCL8 which can contribute to intestinal inflammation in CD and UC. Additionally, neutrophils can release reactive oxygen species (ROS) that damage intestinal tissue and exacerbate inflammation [10,21]. On the other hand, neutrophils may also play a protective role by promoting tissue repair and wound healing: for example, CD177+ neutrophils, as a functionally activated subset of neutrophils, have been found to play a protective role in regulating intestinal mucosal inflammation through increased bactericidal activities, high levels of IL-22, and less production of proinflammatory cytokines [11].

Additionally, neutrophil extracellular traps (NETs), which are web-like structures composed of DNA, histones, and antimicrobial peptides, have been found in inflamed colon tissue, and IBD patients have higher plasmatic concentrations of NETs [38]. When IBD is active, NETs seem to aggravate colon tissue damage and promote thrombotic propensity of these patients. This evidence suggests that a potential therapeutic approach for the management of IBD may involve strategies directed against NET formation [38].

#### 2.5.2. Eosinophils

Eosinophils are white blood cells that originate in the bone marrow and play a multifaceted role in regulating biological pathways in both health and disease. These cells release a diverse range of bioactive molecules, including cytotoxic proteins, cytokines, and enzymes, contributing to mucosal barrier function and immune regulation [39,40].

While eosinophils are typically associated with allergic reactions and parasitic infections, they have also been found to be present in the colon of IBD patients, when their uncontrolled cytotoxic effector functions cause damage to host tissues [39,40,41]. Studies have shown that mucosal eosinophil levels are elevated in IBD patients, even compared to those with food allergies [41]. This increase is likely due to heightened eotaxin-1 production by colonocytes, macrophages, and B lymphocytes in the lamina propria, further highlighting the role of eosinophils in IBD pathogenesis [39].

Eosinophils can disrupt the gut’s mucosal barrier through several mechanisms. Activated eosinophils, with their extended lifespan in the IBD gut, release cytotoxic granule proteins (ECP) and pro-inflammatory cytokines upon degranulation. These cytotoxic proteins, particularly major basic protein, directly target the gut epithelium, impairing its barrier function. Secondly, eosinophils can indirectly compromise barrier integrity through interactions with mast cells. Cholinergic signals, received by muscarinic receptors on eosinophils, trigger the release of corticotropin-releasing factor, which in turn induces degranulation of nearby mast cells, ultimately increasing mucosal permeability [39].

Beyond their direct effects, eosinophils can orchestrate a cascade of inflammatory events by modulating neutrophil activity. Eosinophils promote a massive influx of neutrophils into the lamina propria and their subsequent migration across the epithelium into the colorectal mucus. This accumulation of both cell types within the mucus layer, a previously underexplored site of immune activity, creates a unique milieu that may contribute to inadequate immune responses in IBD. The combined cytotoxic effects and ETosis exerted by neutrophils and eosinophils on both sides of the colonic epithelial barrier further damage the epithelium, potentially exacerbating ulceration [39].

Thirdly, eosinophils can interact with other immune cells, such as mast cells and T cells, further amplifying the inflammatory response. Eosinophils can also produce cytokines and chemokines that recruit and activate other immune cells (e.g., CXCL8 production could induce neutrophil chemiotaxis), leading to the development of chronic inflammation in the gut. In UC, eosinophil impact in pathogenesis is related to Th2 immune response pattern, while the involvement of these cells in CD pathogenesis is less clear [39]. 

However, the role of eosinophils in IBD is not entirely clear. While some studies suggest a pro-inflammatory role, others point towards a potential protective function. For instance, one study found an association between increased numbers of activated eosinophils and inactive ulcerative colitis. This protective effect may stem from eosinophils’ ability to produce protectin D1, a potent anti-inflammatory and pro-resolving mediator. PD1 is thought to dampen eosinophil activity by suppressing their response to chemoattractants and downregulating adhesion molecules like CCR3, both of which are crucial for eosinophil recruitment to inflammatory sites [41].

The role of eosinophils in IBD prognosis is more nuanced than previously thought. While their presence is often associated with inflammation, emerging evidence suggests that they may serve as a positive prognostic indicator. Patients with eosinophil-predominant inflammation tend to experience better outcomes, highlighting the potential of eosinophil levels as a stratification tool in IBD management. Furthermore, the type of inflammatory response appears to be a critical factor, as neutrophil-predominant inflammation is linked to a higher risk of disease flares compared to eosinophil-predominant or mixed inflammation. These findings underscore the need for personalized treatment approaches tailored to the individual’s inflammatory profile [41].

Different studies proposed eosinophils as a therapeutic target in IBD. In fact, considering that eotaxin receptor CCR3 expression is increased in IBD colonic biopsy samples [39], CCR3 blockade prevented mice from developing eosinophilic ileitis and the related remodelling. This suggests that reducing eosinophilic inflammation in IBD may be therapeutically accomplished by targeting CCR3. Furthermore, an antibody anti-ECP (eosinophil cationic protein) suppressed DSS-induced colitis in rats. This implies that a further potential therapeutic strategy for reducing eosinophilic inflammation in IBD may involve targeting ECP, which is released by activated eosinophils [41]. Another approach is the use of biologic therapies that target specific cytokines, such as interleukin-5 (IL-5), which plays a key role in the development and survival of eosinophils. Considering that IL-5 expression has been shown to be related to an early recurrence of CD, it is conceivable that anti-IL-5 agents (for example mepolizumab and reslizumab) can prevent CD recurrence [41,42].

Nevertheless, even though these results imply that eosinophil targeting may be a promising therapeutic strategy for lowering eosinophilic inflammation in IBD, more research is required to fully understand the security and effectiveness of these strategies in humans.

#### 2.5.3. Basophils

Basophils represent <1% of all blood leukocytes and were originally thought to play a role in allergic reactions by releasing HA and other mediators in response to IgE crosslinking on their high-affinity receptor FcεRI. However, recent studies have suggested that basophils may have additional functions beyond their role in allergic reactions. Basophils could play a role in IBD; in fact, some studies show that basophilia of the peripheral blood was significantly associated with UC [43]. Basophils have been found to accumulate in the inflamed colonic mucosa of patients with IBD, including both CD and UC [44].

While research on basophils in IBD is still developing, evidence suggests a multifaceted role for these immune cells. Basophils appear capable of both promoting and regulating intestinal inflammation, potentially influencing disease progression. Studies show they can stimulate memory Th17 responses in vitro and drive the emergence of specific memory T helper cell subsets (IL-17+ and IL-17+/IFN-γ+) in IBD patients, suggesting a role in amplifying inflammatory responses. Furthermore, their interaction with memory T cells may enhance Th2, Th17, and Th17/Th1 effector responses, potentially exacerbating inflammation and contributing to disease flares. Conversely, basophils also exhibit regulatory functions. They express AREG, a cytokine known to enhance the function of regulatory T cells, which suppress excessive immune responses. Additionally, their interaction with innate immune cells involved in IgE-mediated allergic inflammation, such as innate lymphoid cells, hints at a potential role in modulating allergic responses in the gut. The increased presence of basophils in IBD could also reflect a negative feedback mechanism attempting to dampen inflammation [44]. This suggests that basophils may contribute to the pathogenesis of these diseases by influencing pathogenic T cell responses in tissues.

Colonic basophil infiltration was significantly increased in patients with UC, and these seems to be related with their clinical manifestation; in fact, in murine OXA-induced colitis, a significant increase in basophil infiltration was observed, and when basophils were depleted by diphtheria toxin, inflammation improved significantly and mRNA expression of some proinflammatory cytokines, including TNF-α and IFN-γ, decreased significantly [45].

## 3. Ca^2+^ Signalling in Innate Immune Cells: Potential Therapeutic Targets for IBD

Ca^2+^ signalling plays a critical role in regulating a wide array of immune cell functions, many of which are implicated in the pathogenesis of IBD [10]. In particular, calcium ions (Ca^2+^) act as key messengers within T cells, macrophages, and neutrophils, orchestrating processes such as leukocyte adhesion, migration, activation, cytokine production, and cell survival [19,20]. Dysregulation of these calcium-dependent processes in IBD can modulate the production of pro-inflammatory cytokines and the migration of immune cells to the gut. Therefore, understanding the intricacies of Ca^2+^ signalling in immune cells holds significant potential for identifying novel therapeutic targets for IBD treatment. Table 1 summarize the potential involvement of immune Ca^2+^-dependent processes in IBD.

### 3.1. Monocytes

Calcium plays a role in inflammation in monocytes. The calcium-sensing receptor (CaSR), expressed in human monocytes, is a G protein-coupled cell-surface receptor that enables the cell to respond to small changes in the extracellular ionized calcium concentration [16]. The CaSR seems to be implicated in various inflammatory diseases: Two independent studies have highlighted the crucial role of calcium in triggering inflammatory responses. Both studies demonstrated that calcium can stimulate the secretion of the pro-inflammatory cytokine IL-1β from monocytes. This effect was mediated by the activation of the NLRP3 inflammasome, a key component of the innate immune system. Importantly, both studies identified the calcium-sensing receptor as a critical mediator of this process, with NLRP3 inflammasome activation being dependent on both CaSR expression and activity [16]. Furthermore, some studies suggest that CaSR could modulate both the intestinal inflammation and barrier function, which are key features of IBD; for example, Elajnaf et al. found out that the activation of CaSR by its agonist cinacalcet reduced inflammation and improved barrier function in a mouse model of colitis [46]. Cheng et al. demonstrated the crucial role of the calcium-sensing receptor in maintaining intestinal health using a mouse model of colitis. Their study found that mice lacking CaSR specifically in their intestinal epithelial cells were significantly more susceptible to dextran sulphate sodium (DSS)-induced colitis. This increased susceptibility was marked by several key findings: a heightened influx of immune cells into the bowel, elevated levels of pro-inflammatory cytokines, disruptions in the gut microbiome composition, and a compromised intestinal epithelial barrier. This barrier dysfunction was evidenced by the reduced expression of tight junction proteins like claudin-2, highlighting the importance of CaSR in regulating intestinal permeability and, consequently, susceptibility to inflammatory insults [16].

Ca^2+^ signalling in immune cells is partly regulated by TRP (transient receptor potential) channels, non-selective cation channels that are permeable to calcium ions (Ca^2+^). TRP channel members showed variable mRNA expression levels in PBMCs (peripheral blood mononuclear cells) from IBD patients, which may be crucial for the development of the disease. It has been shown that TRPV2 and TRPC1 mRNA expression was lower in PBMCs of UC and CD patients compared to healthy controls, but TRPM2 mRNA expression was higher. Furthermore, CD patients had lower TRPV3 and higher TRPV4 mRNA expression. Moreover, individuals with CD had higher levels of TRPC6 mRNA expression than individuals with UC. Additionally, there was a propensity for TRPV2 mRNA expression to be negatively correlated with disease activity in both UC and CD patients, whereas TRPM4 mRNA expression was only found to be negatively correlated with disease activity in UC patients [18].

Therefore, it is suggested that CaSR and TRP channel members in these cells play a role in regulating inflammation and maintaining intestinal homeostasis, which could be relevant to IBD. However, further research is needed to fully understand their roles and functions in these diseases.

### 3.2. Macrophages

As well as in almost all types of cells, macrophages’ activation processes can also be Ca^2+^-mediated [28,47]. TRP channels, non-selective cation channels that are permeable to calcium ions (Ca^2+^), have been shown to play a role in macrophage function and inflammation. Calcium influx through TRP channels can activate downstream signalling pathways that contribute to macrophage responses, such as the production of pro-inflammatory cytokines and chemokines. Particularly, the transient receptor potential canonical (TRPC) channels seem to play a role in gut inflammation. There is evidence with regards to the effects of TRPC on the intestine: some scientists demonstrated that lacking all of the seven TRPC proteins, in a mouse colitis model, increased infiltration of monocytes and neutrophils in the intestine and promoted M1 macrophage polarization, inducing pro-inflammatory factor secretion. Considering that non-steroidal anti-inflammatory drugs (NSAIDs) block TRPC channels, these results suggest that NSAID intake may aggravate the development of colitis [17].

### 3.3. Dendritic Cells

There are no data on Ca^2+^ signalling’s involvement in DCs in the context of IBD pathogenesis. Just in one study, in which an enrichment of CD11c+ DCs in the lamina propria (LP) of UC and CD patients was found, has it been reported that the function of these immune cells is regulated by store-operated calcium entry (SOCE), which results from the opening of calcium release-activated calcium (CRAC) channels formed by ORAI and STIM proteins [10]. However, it is known that Ca^2+^ signalling plays a critical role in various aspects of DC function, including antigen presentation, cytokine production, and migration. During antigen presentation, DCs take up antigens from their environment and process them into small peptides that are presented on the surface of the DCs in the context of major histocompatibility complex (MHC) molecules: the presentation of these peptides to T cells is a critical step in the initiation of the adaptive immune response. Ca^2+^ signalling is involved in the activation of various signalling pathways that lead to the expression of MHC molecules and co-stimulatory molecules on the surface of DCs. Ca^2+^ signalling can activate the transcription factor NFAT, which in turn regulates the expression of genes involved in MHC and co-stimulatory molecule expression. In addition, Ca^2+^ signalling can regulate the activity of proteases and other enzymes involved in antigen processing, as well as the trafficking of MHC molecules to the cell surface [48]. Ca^2+^ signalling plays a critical role also in cytokine production. Ca^2+^-dependent activation of NF-κB regulates the expression of a large variety of cytokines in DCs, such as IL-12 and TNF-α [48].

Although Ca^2+^ signalling has been found to play a role in antigen presentation by DCs, and consequently in the initiation and regulation of the adaptive immune response, as well as in cytokines production, further research is needed to value this signalling as a potential therapeutic target in IBD.

### 3.4. Mast Cells

Calcium signalling plays a critical role in MC activation and the release of pro-inflammatory mediators [49,50]. Upon activation, MCs undergo a rapid influx of calcium ions (Ca^2+^) from the extracellular space into the cytoplasm, which triggers a series of downstream signalling events that lead to the release of histamine, cytokines, and other mediators. In addition, calcium influx can activate downstream transcription factors that regulate gene expression and promote mast cell survival and proliferation. Dysregulation of calcium signalling has been implicated in the pathogenesis of various mast cell-related diseases, such as asthma and allergic rhinitis. Therefore, targeting calcium signalling pathways in MCs could be an emerging therapeutic strategy for modulating inflammatory responses in these diseases [28,49]. However, there are no studies focused on mast cell Ca^2+^ signalling in IBD; only one paper has it been reported that MRGPRX2 (a mast cell predominant G-protein-coupled receptor) activation leads to calcium mobilization in mast cells, which is a key step in MC degranulation and release of inflammatory mediators, which can contribute to inflammation in UC, suggesting that blocking MRGPRX2 signalling could be a potential therapeutic strategy for treating UC by reducing MC activation and inflammation [51].

As mentioned above, by blocking specific integrin subtypes on mast cells, it may be possible to disrupt their recruitment and activation in the context of IBD inflammation [33]. Considering that integrin activation and binding are generally calcium-dependent processes [52], interfering with calcium signalling pathways crucial for integrin activation could interfere with their recruitment and activation. However, further research is needed to explore this intricate relationship and evaluate the efficacy and safety of targeting mast cell integrins through calcium modulation.

### 3.5. Neutrophils

Neutrophils are key drivers of inflammation and tissue damage in IBD, releasing pro-inflammatory mediators and forming NETs [53]. NETs can further exacerbate inflammation and contribute to thrombosis in the gut [38]. The NET production phenomenon is a calcium-dependent process: calcium activates NADPH oxidase resulting in the production of ROS, and the citrullination of histone H3 (characteristic of NETs) is catalysed by PADs, calcium-dependent proteins [54,55], highlighting the key role of Ca^2+^ in the cellular processes behind the pathogenesis of IBD.

S100A12, a calcium-binding protein secreted by activated neutrophils, has proinflammatory properties and interacts with the multiligand receptor for advanced glycation end products (RAGE) and triggers inflammation via an intracellular nuclear factor κB (NFκB)-dependent signalling cascade. These proteins are strongly expressed in inflamed tissue of patients with active CD and UC, but they are also present in elevated concentration in IBD serum, providing evidence that neutrophils not only play a role within the local mucosal immune system but are also important in systemic immune responses in IBD. In murine models of colitis, blocking agents for RAGE and S100A12 have shown anti-inflammatory effects [56]. These findings suggest that targeting RAGE and S100A12 may provide a new approach to treating IBD. However, further research is needed to determine the safety and efficacy of blocking agents for RAGE and S100A12 in treating IBD in humans.

### 3.6. Eosinophils

While the Ca^2+^-dependence of eosinophil-driven inflammation in IBD requires further investigation, the physiological process of eosinophil activation is well characterized. Circulating eosinophils are primed by agents including IL-5, facilitating their adhesion to the vessel wall. Subsequent exposure to chemoattractants, such as eotaxin, triggers extravasation. The markedly increased shear stress produced during the rolling phase triggers the perfusion-induced calcium response (PICR), which in turn commits the cells to adhere strongly and accelerate extravasation [57]. Therefore, it is possible that calcium signalling pathways play a role in eosinophil activation by IL-5, but this has not explicitly been shown.

In addition, calcium seems to play a critical role in eosinophil migration and actin reorganization; in fact, the inhibition of calcium signalling pathways significantly hampered the capacity of eosinophils to move across fibronectin-coated surfaces [57]; considering that fibronectin protein is abundantly expressed by surface colon epithelial cells [58], it is possible that similar mechanisms may be involved in eosinophil migration in the inflamed intestine.

### 3.7. Basophils

There are no data on Ca^2+^-dependent basophil processes being involved in IBD, but it is well known that calcium signalling is fundamental for many cell functions; in fact, recent studies show that calcium plays a role in the inflammation processes of basophils. When the high-affinity receptor for IgE, FcεRI, is activated by antigenic crosslinking of bound IgE, it results in calcium fluxes and protein kinase cascades driving an array of functional proinflammatory responses in basophils. These responses include the release of histamine, leukotrienes, and cytokines, which contribute to the inflammatory response [59].

## 4. Concluding Remarks

The purpose of this review is to highlight both the critical function of innate immunity cells in the pathogenesis of IBD and the role of Ca^2+^ dysregulation as a potential therapeutic target in IBD patients (Table 1). Some scientists have explored the use of calcium channel blockers as a potential treatment for IBD [60,61]. Immune cells play a dual role in UC and CD: on the one hand, they contribute to inflammation and tissue damage; on the other hand, they have a protective role in the gut. Considering that these processes are usually Ca^2+^-dependent, it would be necessary to deepen this field in order to find new therapeutic targets for IBD treatment.

## Figures and Tables

**Figure 1 cells-13-01079-f001:**
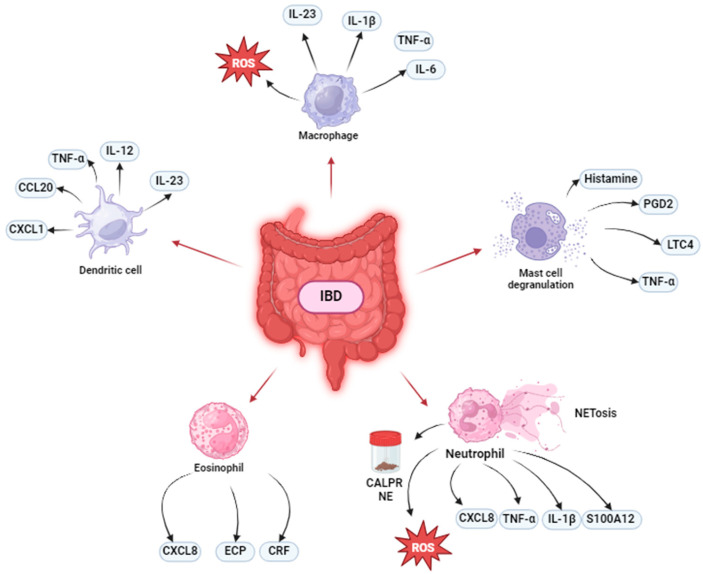
Innate immune cell involvement in IBD pathogenesis. In IBD, macrophages secrete pro-inflammatory mediators, such as TNF-α, IL-1β, IL-6, IL-23; dendritic cells release cytokines (IL-12, IL-23, TNF-α) and chemokines (CCL20, CXCL1); neutrophils are the main producer of CALPR, a gold-standard marker for predicting active IBD, and NE, which could be a novel marker of active disease. Neutrophils also produce cytokines/chemokines (TNF-α, IL-1β, and CXCL8), ROS, S100A12, and NETs. Furthermore, in IBD, eosinophils stimulate immune cell recruitment and produce ECP and CRF and mast cells release Histamine, PGD2, LTC4, and TNF-α. All these mechanisms can be related to IBD pathogenesis and symptoms.

**Table 1 cells-13-01079-t001:** Overview table of innate immunity cell involvement and the role of Ca^2+^ dysregulation in the pathogenesis of IBD.

Cell Type	Pathogenesis	Ca^2+^-Involvement	Therapeutic Approach
Monocytes	**↑** CD16− monocytes **↓** CD16+ monocytes	CaSR activation reduced inflammation and improved barrier function; Lacking CaSR induce higher recruitment of immune cells, higher expression of pro-inflammatory cytokines, reduced expression of tight junction markers.	
Macrophages	TNF—α IL-1β IL-6 IL-23/IFN-γ axis activation Increase of Th17-inducing activity	TRPC activity: (a) Lacking TRPC proteins promoted M1 (inflammatory) polarization; (b) Genetic deletion or pharmacological inhibition of TRPC1 reduced inflammation and improved barrier function.	TNF-α blockade. a4b7 blockade. Thiopurines. IL-4. Arginase. Polyamines. SCFAs.
Dendritic cells	IL-12. IL-23 (Th17 cell differentiation, IL-17A and IL-17F production). TNF-α. CCL20 CXCL1 Gut microflora hyperresponsiveness.	Ca^2+^ signaling-dependent NF-κB activation regulate cytokines production; Ca^2+^ signaling-dependent NFAT activation is involved in antigen presentation; Ca^2+^ signaling regulate the activity of enzymes involved in antigen processing.	
Mast cells	Histamine PGD2 LTs (e.g., LTC4) TNF-α	MRGPRX2 activation leads to Ca^2+^ mobilization in MCs and their degranulation and release of proinflammatory mediators.	MC stabilizers (cromolyn sodium, ketitofen). Activation of LIMR3. Protease, tryptase and histamine inhibitors. a4β7 integrin blockade. Administration of S. cerevisiae-derived β-glucan.
Neutrophils	Faecal CALPR and NE TNF-α CXCL8 IL-1β ROS NETs S100A12	NETs production is Ca^2+^-dependent.	RAGE blockade. S100A12 blockade.
Eosinophils	ECP CRF production induced MC degranulation Immune cells recruitment		CCR3 blockade. ECP blockade. IL-5 blockade.
Basophils	TNF-α. Promotion of emergence memory IL-17+, IL-17+/IFN-γ+ Th cells. Induction of Th2, Th17, and Th17/Th1 effector responses.	FcεRI activation results in Ca^2+^ fluxes with consequently basophil release of HA, LTs, and cytokines.	

CaSR: calcium-sensing receptor; TRPC: transient receptor potential channel; NFAT: nuclear factor of activated T-cells; MRGPRX2: mas-related G-protein-coupled receptor member X2; RAGE: receptor for advanced glycation endproducts.

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
