# Peer review of "Ca2+-Dependent Processes of Innate Immunity in IBD"

_cells, 2024, doi:10.3390/cells13131079_

Round 1
Reviewer 1 Report
Comments and Suggestions for Authors
The authors have done a good job of recapitulating the different innate immune cells involved in IBD, and the potential role of calcium is discussed here. This reviewer finds two different points that will improve the manuscript:
Include more recent work on IBD (the most recent article noticed in the references section was published in 2022); gut microbiome is relevant here.
Specifically include a section on anti-adhesive therapies, which are mainly based on blocking antibodies against integrins (which in turn depend on calcium for effective adhesion) This can be part of the introduction (line 55), and clinical trials can be also included here, together with other biological therapies.
Minor:
Some editing is required (line 145, rephrase line 190, rephrase line 481, etc.)
Bibliography needs revision: there are some articles without page numbers.
Comments on the Quality of English LanguageThe English is readable. Some expressions are not in keeping with the academic tone of the Journal.
Author Response
Dear Reviewer,
Please find attached the detailed responses.

Reviewer 2 Report
Comments and Suggestions for Authors
The manuscript by Palestra et al. reviewed the literature on Ca2+ signalling-mediated processes of innate immunity in inflammatory bowel disease (IBD). Ca2+ signals play a crucial role in immune system function such as in cytokine and chemokines secretion, therefore, the authors discuss the potential involvement of immune cells and their Ca2+-dependent processes in IBD pathogenesis and the possibility to modulate these signallings as therapeutic way. The authors discussed Ca2+ signaling in monocytes, macrophages, dendritic cell, mast cells, neutrophils, eosinophils, and basophils.This review is well performed and interested. However, there are some minor issues that should be considered when revising the manuscript.
1. The authors are suggested to draw a summary diagram.
2. The authors are suggested to discuss the therapeutic approach (Table 1) in a separate paragraph.
3. Whether calcium channels inhibitors or calcium signaling inhibitors play a role? Whether there are preclinical applications?
Author Response

(The authors gave the same response as above.)
